# Joining Sustainable Design and Internet of Things Technologies on Campus: The IPVC Smartbottle Practical Case

Ana Filomena Curralo [1,2,*], Sérgio Ivan Lopes [1,3,4], João Mendes [1] and António Curado [1,5]

1 Escola Superior de Tecnologia e Gestão, Instituto Politécnico de Viana do Castelo, Rua da Escola Industrial e Comercial de Nun'Alvares, 4900-347 Viana do Castelo, Portugal; sil@estg.ipvc.pt (S.I.L.); jmiguelmendes@ipvc.pt (J.M.); acurado@estg.ipvc.pt (A.C.)
2 ID+—Instituto de Investigação em Design, Media e Cultura, 3810-193 Aveiro, Portugal
3 ADiT-LAB, Instituto Politécnico de Viana do Castelo, Rua Escola Industrial e Comercial Nun'Álvares, 4900-347 Viana do Castelo, Portugal
4 IT—Instituto de Telecomunicações, Campus Universitário de Santiago, 3810-193 Aveiro, Portugal
5 ProMetheus, Instituto Politécnico de Viana do Castelo, Rua da Escola Industrial e Comercial de Nun'Alvares, 4900-347 Viana do Castelo, Portugal
* Correspondence: anacurralo@estg.ipvc.pt

**Abstract:** Higher education institutions (HEIs) are favored environments for the implementation of technological solutions that accelerate the generation of smart campi, given the dynamic ecosystem they create based on the involvement of inspired and motivated human resources (students, professors, and researchers), moving around in an atmosphere of advanced digital infrastructures and services. Moreover, HEIs have, in their mission, not only the creation of integrated knowledge through Research and Development (R&D) activities but also solving societal problems that address the academic community expectations concerning environmental issues, contributing, therefore, towards a greener society embodied within the United Nations (UN) Sustainable Development Goals (SDGs). This article addresses the design and implementation of a Smartbottle Ecosystem in which an interactive and reusable water bottle communicates with an intelligent water refill station, both integrated by the Internet of Things (IoT) and Information and Communications Technologies (ICT), to eliminate the use of single-use plastic water bottles in the premises of the Polytechnical Institute of Viana do Castelo (IPVC), an HEI with nearly 6000 students. Three main contributions were identified in this research: (i) the proposal of a novel methodology based on the association of Design Thinking and Participatory Design as the basis for Sustainable Design; (ii) the design and development of an IoT-enabled smartbottle prototype; and (iii) the usability evaluation of the proposed prototype. The adopted methodology is rooted in Design Thinking and mixes it with a Participatory Design approach, including the end-user opinion throughout the Smartbottle Ecosystem design process, not only for the product design requirements but also for its specification. By promoting a participatory solution tailored to the IPVC academic community, recycled plastic has been identified as the preferential material and a marine mammal was selected for the smartbottle shape, in the process of developing a solution to replace the single-use plastic bottles.

**Keywords:** sustainable design; smart campus; smartbottle; participatory design; design thinking; RFID; internet of things; plastic waste reduction; customer-focused technology

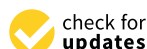



## 1. Introduction

Sustainable development is expected to foster a more compassionate, fairer, and broad-minded exploration of planetary resources, strongly committed to reinforcing the respect for life on Earth for future generations, by respecting mankind and nature. This embodies the basis of the AGENDA 2030 to reach the United Nations (UN) Sustainable Development Goals (SDGs) [1]. To reach more sustainable development, Higher Education Institutions (HEIs) must create integrated knowledge throughout research and investigation

activities focused on solving-problem projects designed to reach academic community expectations [2–5]. Additionally, HEIs play a leading role in the educational training of their public, therefore assuming a major intervention in Sustainable Development [6–8]. The Polytechnic Institute of Viana do Castelo (IPVC), in northern Portugal, is such an institution that is fully committed to playing a central role in the implementation of policies that contribute to a circular, low-carbon economy and sustainable socio-environmental systems in close alignment with regional, national, cross-border, and global strategic options, namely those defined in the UN SDGs.

Under the scope of sustainable development, IPVC is conducting a pilot project named Refill_H2O, funded by EEA Grants Portugal, that aims to eliminate the use of plastic water bottles on the IPVC Campus and therefore contributes to the UN SDGs #4 and #14, "Quality Education" and "Life Below water, respectively, through the design and development of an interactive smart and sustainable bottle that communicates with an intelligent water refill station to promote ecologically correct attitudes among local users, such as students, professors, researchers, and other academic staff, thus contributing to the reduction of plastic consumption in bars, canteens, and residences within the IPVC premises. The final objective of this project is to tackle plastic pollution in the Earth's environment, which adversely affects wildlife, wildlife habitats, and humans, contributing to the achievement of the UN SDG #14. Although all marine life is in decline due to ocean pollution, in the next 30 to 50 years, a large proportion of marine animals could lose more than half of their population due to hazardous substances in seawater.

By embracing the Refill_H2O project, the IPVC desires to take a lead as a sustainability model of excellence [5,9–11], therefore bearing special responsibilities concerning environmental sustainability and sustainable development, understood as the ability to meet present needs without compromising future generations' ability to meet their needs. Additionally, Refill_H2O plans to encourage ecological behaviors in the academic community contributing to a paradigm shift through new habits by favoring the eradication of single-use plastic water bottles and respective waste and pollution. The project involved staff, students, teachers, and researchers, demonstrating through real-life problem-solving the meaning of sustainable design and customer-focused technology.

The in situ implementation of the Refill_H2O project reaches all of the IPVC premises, including bars, canteens, and residences. In addition to raising ecological awareness and change towards a sustainability mindset, the tangible system developed is composed of a sustainable reusable smartbottle that interacts/communicates with a refill station supplying filtered water. Both products (smart, sustainable bottle and water refill station) were subject to a previous survey applied throughout the IPVC community based on the methodological concept of Design Thinking towards the coexistence of social and technological development options in systems that require human interaction, involving the users in the design of the products they will afterward use [12]. The development of both products progressed simultaneously since the implementation of interactive artifacts poses technological challenges, namely concerning the identification of technologies for wireless communication which must allow the interaction between the bottle and the refill station.

This interaction is supported by the application of Information and Communication Technologies (ICT) and the Internet of Things (IoT), exploring the use of short-range radiofrequency communications to allow greater interoperability between the smart and sustainable bottle and the refill station. For that, the bottle design includes a radio frequency identification (RFID) chip for easy integration with the refill station management system, allowing an automatic recharge process without physical contact with the equipment. Additionally, the refill station management system provides a set of indicators to allow the assessment of quantifiable sustainability parameters, such as the estimated amount of avoided plastic waste and energy savings resulting from the global reduction of waste, the reduction of greenhouse gas (GHG) emissions, and information on the environmental footprint of each system user. The development of these products (smart, sustainable bottle

and water refill station) was only possible due to the interaction between the disciplines of Design and Engineering, which could provide an integrated design solution contributing to the evolution and improvement of the final products, by considering the specific needs of users, as inspired by Brown's Design Thinking models [13]; a scenario where ICT and IoT technologies play a fundamental role, encouraging the autonomy of users, and pedagogically allowing them to recognize, identify and reduce the environmental footprint. In addition, the association between sustainable design and technological innovation plays a decisive role in the change of behavior towards sustainable practices, promoting ecoliteracy and systemic changes both inside and outside the IPVC premises. Design Thinking allowed the use of problem-solving methods that respond to the needs of individuals in a technological way [14].

This article addresses the design and implementation of a smartbottle that communicates with an exclusive drinking water dispensing system, designed to enhance the final users' enthusiasm and motivation towards environmentally friendly approaches, considering nature resources and more planet-friendly materials as part of the design process, to eliminate single-use water plastic bottles purchased in the IPVC premises, involving almost 6000 students, 51,000 small plastic bottles (0.5 L) and 15,000 large plastic bottles (1.5 L), translating into approximately 1215 kg of plastic waste averted. To stimulate the sustainability mindset and ecological awareness, the water refill station was designed to display information concerning individual water intake but also environmental sustainability metrics and indicators, such as the estimated amount of averted plastic waste, the energy-saving from overall waste reduction and the reduction of greenhouse gas emission, and information on the user's environmental footprint.

These elements are the reason why the communication aspect of the system was developed. Besides motivating group interaction and integration through belonging to the exclusive drinking water system, this system is designed to be pedagogical and to encourage sustainability concerns and further action, beyond the boundaries of this specific sustainable design project and beyond the walls of a higher education institution. A durable design solution intended as a reminder of a common goal to reach zero waste, it includes the shape of a marine mammal as a personification and reminder of the reason why reducing, reusing, and recycling are important for individuals, communities, and the environment while saving money, energy, and natural resources. As a result, three main contributions have been put forward: (i) the proposal of a novel methodology based on the association of Design Thinking and Participatory Design as the basis for Sustainable Design; (ii) the design and development of an IoT-enabled smartbottle prototype; and (iii) the usability evaluation of the proposed prototype.

In addition to the development and implementation of an innovative system based on the long-term use of recycled materials and information technologies, the Refill_H20 challenge guarantees good quality of the stored water and involves the whole academic community, including staff, teachers, and almost 6000 students, most of which are young adults and more prone to mindset changes. The project offers a reflection topic, a subject for discussion and careful consideration, and provides substantial information on sustainability and change in mindset and attitudes. The elimination of disposable plastic bottles at the gym and the increase of water intake as an essential habit to promote a healthy diet, and the recognition of water as essential to life on the planet, affect all species.

In short, this research aims to respond to sustainability challenges at the academic level, namely by promoting the elimination of single-use plastic bottles in bars, canteens, and halls of residences within the IPVC premises, thus enabling achieving two major objectives: (1) to meet UN SDG #14, which is related to the conservation and sustainable use of the oceans, seas, and marine resources; and (2) by promoting change in the mindset and attitudes in the academic community, therefore helping to meet the UN SDG #4, aiming to ensure inclusive and equitable quality education to encourage sustainability concerns and further action among the academic community. To this end, a survey amongst the IPVC academic community has been performed to involve end-users in the design



process of the products they will afterward use. The results have been analyzed and used for the specification of the IPVC Smartbottle Ecosystem, which comprises an interactive smartbottle that communicates with an intelligent water refill station. The overall ecosystem was designed and developed to promote ecologically proper attitudes amongst the IPVC community, thus contributing to the reduction of plastic consumption in the academy. The impact of using such a system on the academic community is still under monitoring.

This document is organized as follows: Section 2 introduces background theoretical concepts; Section 3 presents the Materials and Methods, based on the Design Thinking methodology by using the participatory design approach; Section 4 presents the results of the participatory design process and its related surveys, the prototyping stage, and the technological approach; the discussion of results in Section 5 identifies a transdisciplinary approach, joining different areas such as Product Design, Electronics, and Materials Engineering; finally, the conclusions present the main achievements of the present research.

## 2. Background

### 2.1. Sustainability and Design Thinking

Over-exploitation of natural resources, massive consumerism, and the excess of existing products and respective waste and pollution have worldwide effects. Particularly plastic pollution, being such a persistent material, has a long-term ecological, economic, and eco-toxicological effect. Information and mindset changes are key for a sustainable future. Ecoliteracy is one of the main agents of change in a feedback process between society and industry, towards sustainable manufacturing, minimizing negative environmental impacts while conserving energy, and natural resources over their whole life cycle, from the extraction of raw materials until the final disposal.

The expression 'Sustainable Design' refers to a rational, structured process to create something new [15] to solve problems concerning sustainability [16]. The sustainable design emerged in the 1960s along with the concept of sustainable development. At the time, the visionary American architect and philosopher Richard Buckminster Fuller declared that a comprehensive anticipatory design science should be adopted to create an operational manual for spacecraft Earth in order to guide human development while preserving the environment, optimizing the use of resources, and ensuring their fair distribution [17]. In the 1970s, Victor Papanek developed these ideas in his book Design for the Real World [18], which may be considered the steppingstone for the theory of sustainable design [19].

Currently, sustainable design is implicated in ecoliteracy and in the environmental and social impacts this project will have on a restricted community and the world. Acting as a philosophy, the sustainable design integrates an environmentally friendly approach and considers nature resources and more planet-friendly materials as part of the design. Inviting the system users to reuse more, recycle more, and reduce more, because reducing, reusing, and recycling can help individuals, communities, and the environment, saving money, energy, and natural resources. This holistic approach thus combines environmentally responsible design and social responsibility [20].

About 80% of a product's environmental impact is defined in the early design stages of product development [21]. Designers are responsible for specifying the material compositions of products, how the raw materials are processed or formed (manufactured), and how products are packaged, distributed, used (to some extent), and eventually disposed of. Every decision made during the design of a product or product-service system will have a direct social and environmental impact (negative or positive) on people and our planet. Sustainable product design is situated in the context of the growing concern about the degradation of ecosystems and the availability of resources for future generations [22]. Hence, design thinking principles, such as user focus, have led to the identification and incorporation of relevant user needs and behaviors toward the development of product sustainability. In the context of social innovation, the authors Brown and Wyatt [23] maintain that design thinking addresses the needs of the people who will consume a product or service and the infrastructure that enables it. Design thinking is an adaptive and iterative

process that contrasts with a rigid set of methods. Instead, design thinking guides teams through a recursive process, using various tools within an overall design philosophy [24].

To create sustainable value, the Refill_H20 project includes all target users, fostering a wider ecological awareness, improving the overall ecoliteracy and enhancing a sustainability mindset, especially among young design students. For this purpose, when enrolling for the first time, the students can purchase a smartbottle, which will accompany the student throughout their whole academic journey. The sense of belonging to a community is broadened by the shape of the bottle, which personifies a large marine mammal, an Orca, symbolizing the far reach of individual actions. Life on earth, in general, is threatened by human action, waste, and pollution. One single animal was chosen to represent the collective will to change that for a sustainable future [22,25]. In this project, design thinking is regarded as a problem-solving approach for designers to integrate the specifications of end-users and key stakeholders throughout the solution development process [24].

The development of an intelligent, reusable bottle, called the smart and sustainable bottle, with innovative and sustainable characteristics, designed to communicate with a technologically advanced refill station for the supply of filtered water, were both defined after consultation with the academic community of IPVC, applying a survey inspired by the methodological concept of Design Thinking, divided into stages as shown in Figure 1.

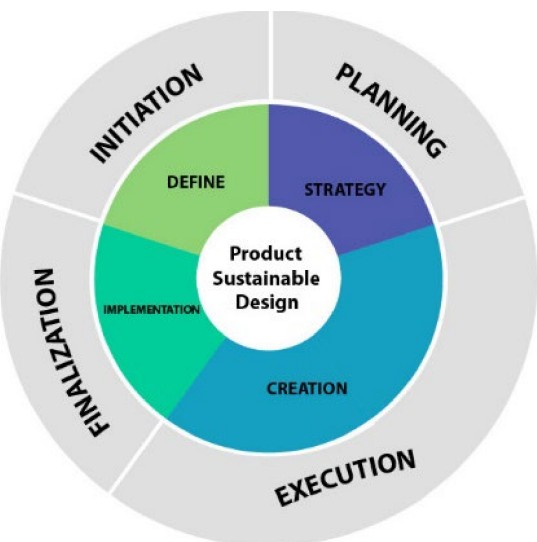

**Figure 1.** Product-sustainable design model. Adapted from Brown [13].

The systemic model of circular design presented in Figure 1 is structured in four interconnected layers representing sustainable product development. This model reinterprets Brown's 2009 thinking model [13], applied in a circular perspective in order to holistically integrate circularity considerations, tools, and methodologies as central activities in the development of new and efficient products, systems, or services. Design thinking is particularly useful in solving comprehensive sustainability-related problems, as it explores the context of the problem before mapping the scope of innovation [26]. Design thinking considers that problem defining depends on the system in which it emerges [27]. Therefore, it takes a systems perspective that does not just focus on the obvious problem but also correlates it to the surrounding system [28].

In this case, that is the reduction of single-use plastic, the user needs, and emerging trends. Thus, it allows a holistic understanding of complex issues related to sustainability and finding non-obvious root causes [29]. The Refill_H2O project process focused mostly on creation and execution, project development, prototyping, and product testing. This process was rather complete since it allowed the integration of different design methods, combining different disciplines. They all contributed to the project's progression according to the

needs at each stage. However, the results at any time can lead to revisiting the previous stages, for example, to reformulate the problem or to involve new stakeholders [30].

Through this model, the designer is continually reflecting on evidentiary facts, principles, and tacit knowledge. Subsequently, judgments take place throughout the whole design process [25]. The designer assumes the role of the main linking agent connecting different areas, such as Materials Engineering and Electronics, adding a semantic value to enrich the user experience and benefit from the rapport between the system (bottle and refilling station) and the user. Thus, a collaborative approach is proposed through design thinking, involving the stakeholders in the design process [31]. This is beneficial mainly for two reasons. Firstly, the underlying notion of participation makes it clear that all people are affected by a specific sustainability issue, and subsequently by the resulting solution, and should therefore be given a voice in the development process [32]. Secondly, design thinking understands that heterogeneous perspectives and skills are valuable resources and assumes that collaboration in multidisciplinary and cross-functional teams will lead to better innovation outcomes [29,33].

Multidisciplinary Design was evidenced through the global, systemic approach concerning design for sustainability since there were sustainability aspects to consider regarding the design of the artifact itself [34]. This is because the design activity seeks not only to understand and address the "what is it" of a situation, but also seeks "what it can be" or "what it should be" in a given situation, in order to improve it—the design rationale [35].

### 2.2. Case Studies Similar to Refill_H20

This section highlights a set of relevant studies already developed concerning plastic bottled water consumption reduction.

Water is susceptible to contamination. While filtration systems can be used to improve the taste and quality of drinking water, they do not offer complete protection against bacterial growth, which becomes even more critical with larger volumes. In recent years there has been a growing output of single-use plastic water bottles and the replacement of more expensive glass bottles by the industry. The direct result is the accumulation of plastic waste, causing an environmental challenge. The research in this field is directed at reducing the amount of discarded plastic. In Portugal, the Pingo Doce supermarket chain has implemented an exclusive service of Filling Fountains dispensing filtered water, using purpose-made reusable bottles. A partnership with ECO, this innovative, sustainable, and affordable service is available in 1, 5, 3, and 6 L format. The campaign considers that the ECO reusable bottles contribute to the preservation of the planet since plastic waste is one of the main pollutants in our oceans. Since 2018, ECO eliminated more than 200 tons of single-use plastic water bottles [36].

The Woosh company, in Miami, FL, USA, provides smart filling stations and water meeting the highest quality standards. Although the water is paid for, it is supposedly cheaper than buying a plastic water bottle, thus favoring user adherence and reducing the amount of plastic waste. It presents a wide variety of typologies such as indoor, outdoor, mobile, and multi-tap, focusing on water treatment and bottle-rinsing technologies [37]. Developing the traditional roman public water fountains in Rome, Italy, the Water Houses [38] is a project by ACEA Group to combine sustainability and innovation. The high-tech water sources also allow recharging smartphones via USB and offer public information through digital screens. For free, the users can get both plain and sparkling drinking water. Focusing on sustainability, hygiene, and sanitization, in Hong Kong, China, the company Urban Spring [39] aims to reduce the consumption of plastic containers, namely carboys and PET bottles by offering water filling stations with a simple user-friendly interface. Permanent or portable, for indoor areas, the filling stations are equipped with a water filter and sensor to assure the water quality and temperature, although it depends on the bring-your-own-bottle culture.

After this brief presentation of different case studies, a SWOT analysis was performed, as shown in Table 1, identifying strengths, weaknesses, opportunities, and threats, to

analyze scenarios (or environments), to identify their implementations, the design process, differences, and similarities concerning the Refill_H2 O System.

**Table 1.** Case studies SWOT analysis.

| | Strengths | Weaknesses | Opportunities | Threats |
|---|---|---|---|---|
| **Eco** | - Bottle with UV filter that protects and preserves water quality and properties<br>- Great positive environmental impact<br>- Innovative and sustainable way of selling drinking water | - Consumers tend to leave the bottle at home<br>- Water is paid and there is only one way to refill<br>- Sells tap water | - Provides filtered water<br>- New way of selling water<br>- Drastic reduction of plastic consumption | - New forms of business<br>- The purification system (sediment filter, activated carbon filter and UV lamp) removes important components from water |
| **Acea** | - USB charging station<br>- Digital screens and information<br>- Regular and carbonated water<br>- Free of charge | - Discarded water easily falls to the ground creating puddle of water and dirt<br>- Interferes with the view in touristic sites | - More water supply points<br>- Tourists and citizens satisfied, since no cost involved<br>- Drastic reduction of plastic consumption | - Large volume, interfering with the installation site<br>- Costly structure |
| **Well** | - Innovative sustainable way of consuming water<br>- Large positive environmental impact<br>- Simplified information<br>- Digital screen<br>- Free company water | - No information in the structure explaining the project<br>- There should be a dispenser of ecological cups | - More water supply points<br>- Provides filtered water<br>- Drastic reduction of plastic waste | - There is no drinking fountain; users tend to leave bottles at home |
| **Woosh** | - Innovative way of consuming water<br>- Multiple options (hot, cold, regular, carbonated)<br>- Great positive environmental impact<br>- Range of mobile refill stations<br>- Digital screen<br>- Using safety sensors and protocols, stations ensure safe water delivery and automatic shutdown (and alert) when water quality is compromised | - Consumers tend to leave the bottle at home<br>- Water is paid for and employs tap water | - Provides filtered water<br>- New way of selling water<br>- Drastic reduction in plastic consumption | - It is mostly a new form of business<br>- Public may not adhere because they are paying more for the same company water |
| **Refill_H2O** | - Great positive environmental impact<br>- Regular and carbonated<br>- Digital screen<br>- Interactive smartbottle that communicates with refill station<br>- Accessible to all users | - Consumers tend to leave the bottle at home<br>- Water is paid for, and tap water is used<br>- There is only one filling amount | - Provides quality filtered water to users<br>- Drastic reduction of plastic consumption<br>- Instill healthy habits of reducing the ecological footprint, especially among young consumers | - Despite the reduced price, the public may not adhere because they are paying for tap water<br>- The purification system (sediment filter, activated carbon filter and UV lamp) removes important water components<br>- Changing operational peak periods and profiles, school breaks |

Adapted from research work on public water dispensers in urban contexts [40].

The examples previously introduced show that sustainability and innovation can be natively combined to enable a cooperative and participative project development strategy, thus promoting the development of technological products without compromising the environmental impact of the implemented solution. Hence, the significance of this work is relevant, since it combines design, technology, and sustainability in a complete ecosystem. Next, the adopted methodology to perform the implementation of the Refill_H20 project is presented.

### 3. Materials and Methods

The proposed model was characterized by a standardized and clear methodology, collaborating not only to teach the design process but particularly towards team cooperation. The initial research defined a systematization of the project supported by a particular methodology, which generated several solutions subsequently evaluated, improved, and developed in a heuristic sequence with a view to meeting the objectives.

Through a multidisciplinary methodology, the designer, in addition to the final development of the product, prioritized the development stages and focused on the target audience and their needs. Simultaneously, the engineer, with a more pragmatic elaboration and directed to the technical result of the final product, determined the problem solutions, results and effectiveness, and the prototype elaboration based on the product design by the designer, taking a different approach in each stage. However, it is essential to emphasize that both designers and engineers go through the same steps, following nevertheless distinct approaches, revealing the search for innovation in research, in the construction of models, testing, redesigning and in the constant search for new solutions. Multidisciplinary teamwork was essential to reach the solutions that responded to the complexity of the task.

As a pedagogical project, the teaching method was to hand over a relatively independent project to the students, oriented by a group of teachers. The project took place in a public HEI, involving teachers and students from different undergraduate courses and disciplines, in a multidisciplinary collaborative project targeting the reduction of plastic bottle waste.

As a problem-solving method, the Design Thinking process is based on the idealization of a solution focusing on the user through an integrative nonlinear approach, in a fundamentally exploratory research method. The process is considered together with the framework conditions, and the problem analysis and solution are systematically and holistically considered in the form of a multistage process [41].

Factors such as time, communication, or complexity compose the context whose interpretation in the design process is a dynamic process, argumentative, cyclical, recyclable, and therefore sustainable. Thinking about proposals in this way is to delve into, idealizing, experimenting, analyzing, and reevaluating products. Supported by a technical engineering concept, the design method proposed in this project is a creative process that recognizes technical issues in addition to ideas and human issues, answering people's needs.

The methodological process requires operation in open development, through advances and setbacks and variations in the current reality. The pragmatic application of this method in engineering becomes a Creative Engineering Design Process incorporating practical characteristics related to the technical execution [42]. In turn, the use of Design Thinking also aims to stimulate creative thinking, improve practices, and project visual thinking [43]. The application of this method allows establishing rules of execution, thus collaborating in better planning and implementation of ideas.

From a methodological point of view, the Refill_H2O project involved an exhaustive survey at the scale of the IPVC (schools, bars, canteens, and halls of residence), to identify water intake habits of the resident population (students, teachers, and staff) concerning plastic water bottles.

## 4. Results

To understand target users, the project considered their actual behaviors, desires, and expectations, as well as their experienced reality. This included a survey application. The survey was determinant for the final product, since it provided the answers to the research questions, informing and enriching the design process. In the survey, the resident population was invited to identify a set of physical, aesthetic, and functional requirements that would become the design specifications for the smartbottle. Divided into three sections, the Consumer, the Bottle, and the Service, the survey collected data to be used as input for the smartbottle design and service quality. The first stage of the participatory design process consisted of immersion, as shown in Figure 2 below.

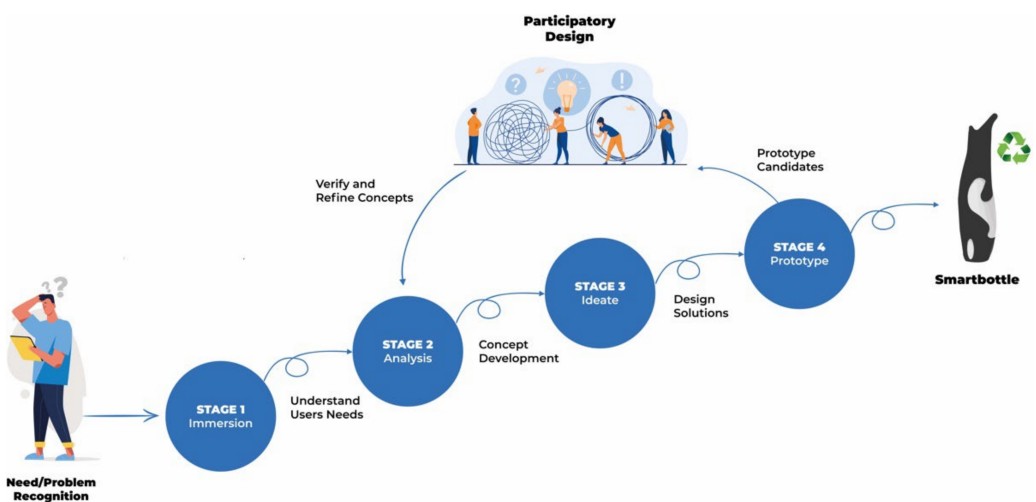

**Figure 2.** Adopted methodology with Participatory Design included.

### 4.1. Survey Results

The first group of questions focused on daily water consumption, preferred locations for regular water collection, and the number of bottles purchased weekly at the institution. The second section focused on the bottle, concerning the key characteristics, such as volume, material, and other relevant factors. The third group of questions is focused on the service, on whether a technological factor associated with the refill station would be appealing if the bottle and the station should be connected by an application, what data was considered relevant to display, the price the user would be willing to pay, payment methods, type of water, if the product was considered useful to reduce the plastic pollution, and if the user would consider using it.

From a pool of 536 respondents, it was possible to identify the gender, age, education level, and occupation, predominantly (80%) students. About 90% of respondents drink water frequently, and more than 90% agree it is useful to monitor their daily water intake. A total of 412 respondents declared using reusable bottles. It was also identified that 96.1% of the respondents prefer a reusable bottle instead of a single-use plastic bottle. Aspects, such as functionality, materials, and cost, were considered the most significant for a smartbottle. The preferred materials were stainless steel, recycled plastic, bamboo, and glass. The distinguishing qualities identified were easy-to-wash, absence of smell or taste in the water, easy to carry, and thermal insulation.

The second stage involved the analysis of survey data, answering three fundamental questions: "What is it?", "When to use it?", "How to apply it?" [42], schematizing and interpreting answers and graphs, crossing all the information collected and deducing relevant considerations to be applied during the design process. A brainstorming session allowed for organizing the collected information, raising new pertinent questions.

Based on the survey results, the material was one of the main issues to address in the reusable bottle design, originating three main keywords: extrusion process, blowing pro-

cess, and injection process, concerning recycled plastic. Another main issue was technology, since the application of electronic devices, such as Radio Frequency Identification (RFID), Bluetooth Low Energy (BLE), and Near Field Communication (NFC) stood out as the most suitable to interface the smartbottle and the refill station. Finally, some important features considered were "visually enjoyable bottle", "easy to wash", "inodorous", "easy to carry", "750 mL" and "low cost". After immersion and analysis, following the interpretation and organization of the survey results, the following steps were ideation and prototyping.

The ideation stage was the moment of total freedom where all proposals and elements were openly considered, with the production of sketches. Several bottles were proposed for selection of the most viable in terms of production, cost, and design. At a conceptual level, sea animals were chosen to represent the wide-scale problem of sea pollution, namely the orca, the seagull, and the sea turtle, and the students produced different sketches for selection, considering the structure of each animal and their respective potential to act as a reminder of the ultimate goal of avoiding plastic waste: to save lives.

### 4.2. Prototyping Results

The result of the sketch selection was a bottle with the shape of an orca. The shape of the bottle was thus based on the physical, morphological traits of this specific mammal, whose immune system is weakened by toxic chemicals, affecting their reproductive capacity [25]. Also, during birth or during the nursing period, parents may transmit pollutants, causing the species to gradually decline.

The purpose of the prototyping stage would be to select the best design in terms of ergonomics, functionality, selected materials, mechanical characteristics, and approach to sustainability issues. The goal of prototyping is not to create a working model. It is to give form to an idea, learn about its strengths and weaknesses, and identify new directions for the next generation of more detailed and refined prototypes. According to Vianna et al. [42], the prototyping process starts with the formulation of questions that must be answered concerning the idealized solutions.

Three different prototypes were budgeted with the proposed materials. Prototype A was a recyclable plastic bottle, highlighting the contrasting colors of the Orcas. Prototype B was a stainless-steel bottle and recyclable plastic stopper, while Prototype C was entirely made of stainless steel as shown in Figure 3.

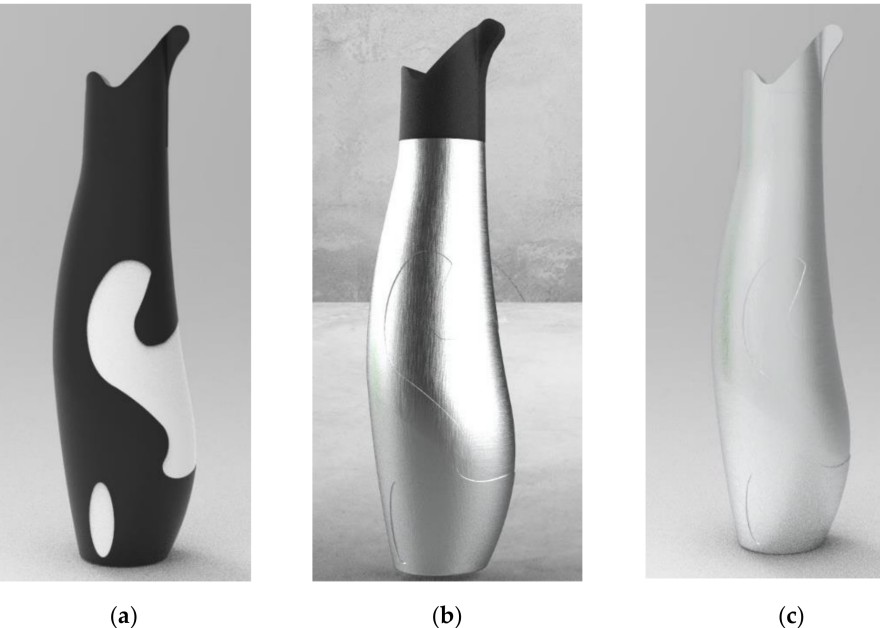

(a)　　　　　　　　　　(b)　　　　　　　　　　(c)

**Figure 3.** Smartbottle prototypes (**a**–**c**).

The curvilinear shape creates an ergonomic handle, facilitating the use and transportation of the bottle. The reliefs on prototypes B and C add friction to the product curves. The bottom projection adds stability to the bottle when placed on a surface and provides the location for the chip that will communicate with the refill station. The following issues were considered concerning the final result of the prototyping process:

1. Since the bottle is asymmetric, aluminum production would require a 5-axis CNC machine.
2. The aluminum solution requires an aluminum casting process with 2 molds (two casts for the body + two casts for the lid).
3. Regarding the recyclable plastic bottle, the chosen material was ABS, as it is the most accessible for the execution of a first prototype,

The prototype was materialized in 3D printing with ABS material in an industrial printer as shown in Figure 4 below.

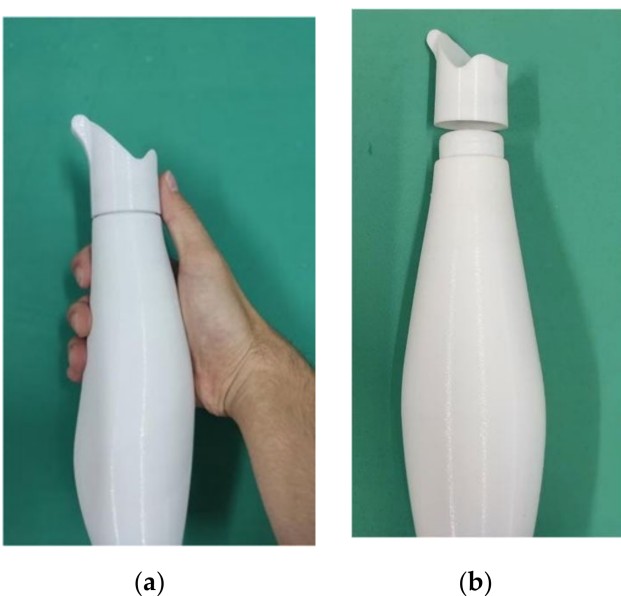

(**a**)                    (**b**)

**Figure 4.** (**a**,**b**) Prototyping.

*4.3. Communication Technology*

Several technologies were considered, such as RFID (Radio Frequency Identification), NFC (Near Field Communication), and BLE (Bluetooth Low Energy). RFID is an identification technology that uses radiofrequency to communicate data and allows a transponder 14 to be read without the need for a direct visual field, through objects made from the most diverse materials, such as wood, plastic, paper, among others. The RFID tag consists of a small object (tag) that can be placed on a person, an animal, or a product [43] having utility in identification, location, and tracking applications. The use of this technology in IoT applications, together with the use of wireless sensor network (WSN) technologies, opens up new possibilities not only in the development of new interactive artifacts, but also in their integration into intelligent services [44]. The use of this technology may have security implications in the specific context of campus [45], however, the advantages of this technology include low cost and reduced tag size, which allows for high scalability and simple tag integration during the production of the interactive artifact [46].

NFC (Near Field Communication): technology aimed at contact communication, progressing from a RFID and Smart Card technologies. This low-range technology operates on a 13.56 MHz frequency, with data transfer up to 424 kbits and with communication initiation when two NFC devices approach. It is physically compatible with RFID tags. This communication technology has been widely used in wearables [47] and traceability applications [48] given the common integration of NFC readers when designing new

smartphones, which enhances the use of technology in the development of new IoT products and applications. However, its operating cost is considerably higher than that of RFID technology.

Bluetooth Low Energy, a technology designed with the purpose of improving the performance of energy consumption of mobile devices [49], such as cell phones, smart watches, and other devices, that are normally used for communications in personal area networks. According to BLUETOOTH SIG15, this is short-range wireless communication technology, with very low energy consumption (ULP–Ultra Low Power), a protocol stack that is lightweight and allows incorporation with existing Bluetooth technology. The main advantages of this technology lie in its low consumption and a communication distance of up to 100 m. However, it is an active communications technology, i.e., it requires a battery, and the tag size is imposed by the type of antenna used [50]. Another disadvantage concerns the final cost of the tag, which is considerably higher than preceding technologies, i.e., RFID and NFC.

As a result, RFID technology was chosen in the smartbottle design, for communications between the smartbottle and the refill station, given the considerably better benefit-cost ratio compared to NFC and BLE technologies, and considering the main advantages of RFID technology, namely low operating cost, small size, and high scalability combined with simple integration during production. All prototypes were designed to include an RFID tag at the bottom for interface with the smart water refill station. Concerning dimensions, the bottle height is 282 mm and 77 mm in width, with a 500 mL capacity. The bottle was structured in two sections, both inspired by the physical and morphological traits of the Orca. The stoppers were inspired by the orca's tail and designed for easy opening.

### 4.4. System Architecture

Figure 5 depicts the operational architecture used within the Refill_H2O application, presenting two use case examples, that represent the interaction between the smartbottle and the refill station. To use the physical water dispenser station, the user must provide authentication via Student ID Card, by placing the card in the RFID tag or via smartbottle, which enables the system to compare the RFID data with the system Smart Water Refill Station embedded database. The RFID reader transmits the user ID, alongside the amount of dispensed water to the Fiware App Server, using the WAN network, allowing a permanent connection between all components, and enabling data management and processing.

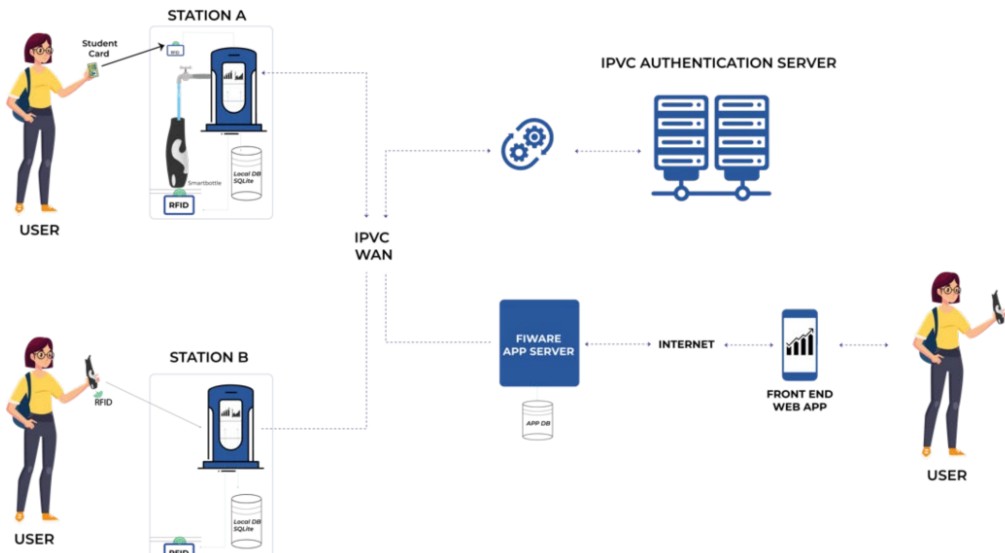

**Figure 5.** Refill_H2O System Architecture.

The proposed operational architecture includes five main components:

1.  Smartbottle (interactive artifact);
2.  the deployed IoT Edge devices (Smart Water Refill Station) that communicate with the Smartbottle through RFID technology and the student identification card for user authentication;
3.  the IPVC Wide Area network, that is, the ICT infrastructure that will perform backhaul communications1;
4.  the IPVC authentication server (which can be accessed in an "as-a-service" approach);
5.  and the FIWARE Application Server, which handles all communication between IoT edge devices, data storage, and client application through a context broker.

To use the refill station, the user must provide valid authentication by placing the smartbottle [51] (p. 21) in the station or inserting an ID card with native RFID technology and placing a conventional water bottle in the station of recharge. The client application is based on responsive web technologies with visual analysis tools and panel-based technologies such as Grafana, presenting a powerful interface to display useful information in a clear and friendly way. The user interface includes three main functional areas: (i) a dashboard that will display relevant metrics (number of recharges per time period, the estimated average amount of water consumption, the estimated amount of plastic waste avoided, energy savings from overall waste reduction, and reduction of greenhouse gas (GHG) emissions); (ii) specific key performance indicators (KPI's) and information on users' environmental footprint; an authentication area allows user authentication, allowing the application to change accordingly; and (iii) a user and system administration area to support back-end operations in relation to user coordination and system administration tasks. This will allow the refill station to be used without the bottle.

The smartbottle integrates with the water refill station, enabling the following features:

*   automatic filling process without physical contact with the equipment;
*   the estimated average amount of water consumption through the client App;
*   number of recharges per period of time for water intake and hydration monitoring;
*   the estimated amount of plastic waste avoided (considering different metrics: temporal, cumulative, individual, or referring to faculties, classes, etc.);
*   energy savings due to the general reduction of waste and greenhouse gas (GHG) emissions;
*   and information about the users' environmental footprint.

When using a smartbottle, the refill station dispenses water up to the bottle's maximum capacity or until the ID card disconnects from the RFID reader. Upon disconnection, the refill station will trigger an event that will store all data in a lightweight, serverless, zero-configuration database engine with no configuration or administration requirements. The intelligent water refill station has a user interface application for a real-time demonstration of different metrics and indicators related to the contribution to waste reduction, reduction of greenhouse gas (GHG) emissions, and other relevant information. Figure 6 shows the refill station with all the main elements identified, namely the refilling zone, the user interface display, the ATM terminal for payments, and the RFID reader.

Gamification is used to promote user motivation and engagement [47–50] by applying game features to a non-game context. This will allow an open competition between schools, selecting who contributes the most to reducing GHG, or who has healthier behaviors regarding water consumption, and the subsequent advantage is the promotion of a cleaner, more sustainable campus.

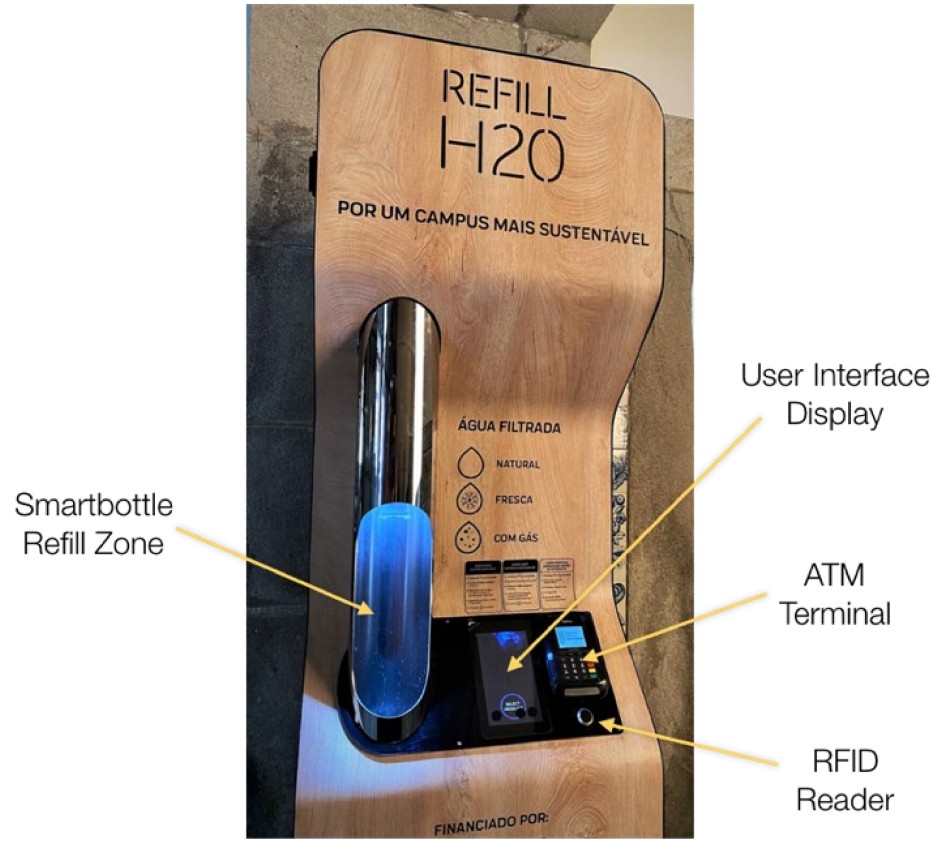

**Figure 6.** Refill Station with identification of main elements.

*4.5. Usability Test Results*

The prototype shown in Figure 3a,b was submitted to usability testing near 102 users to identify and solve problems in order to improve the product's usability. The test evaluated different tasks involved in the smartbottle use, such as picking up, drinking, carrying, and refilling. Prototyping and testing results allowed an understanding of user performance and relationship with the smartbottle and refill station [51].

As a result, 45% of the respondents consider the general impression of product usability as excellent, and 92% consider the size of the prototype adequate, drawing the conclusion that the product would not need resizing adjustments. Given that the main objective of the project is to reduce and prevent users of the IPVC campus from buying conventional plastic bottles, the question was whether after experiencing this product, they would eventually change to plastic bottles. The usability test results show that 91.2% (93 users) would abandon traditional plastic in favor of the smartbottle.

Concerning the materials used in the bottle, it was possible to verify that the presentation of the prototype with 3D modeling graphic elements impacted the final decision of the users. A total of 93.1% (95 respondents) agreed it was the most suitable, while 7 respondents did not agree with this evaluation. There was satisfaction and surprise with the Orca representation and the hygienic and thermal characteristics of the bottle, which were considered more important than elegance or visual and chromatic similarity to the represented animal. 49% of the respondents (50 users) preferred the aluminum bottle.

One of the preponderant features of the initial survey was easy washing. According to the respondents, most of the bottles currently in the market are difficult to wash due to narrow bottlenecks; hence, a larger bottleneck was considered in the product design to allow easy cleaning while keeping good ergonomic functionality (nose not touching the bottleneck while drinking). 87.3% (89 respondents) considered that this product facilitates the cleaning process as it allows cleaning instruments (such as bottle brushes). Bottle

transportation and handling were one of the key requirements for the respondent users, as shown in Table 2.

**Table 2.** Usability questionnaire results concerning requirements.

| Requirements | Percentage | Answer |
|---|---|---|
| Difficult to carry | 94.1% (96 respondents) | Considered the prototype easy to carry |
| Difficult to handle | 99% (101 respondents) | Considered the prototype easy to handle |
| Size of bottleneck | 90.2% (92 respondents) | Considered the bottleneck size of the prototype convenient; however, 9.8% (10 users) prefer smaller bottlenecks due to the risk of the nose touching the bottleneck |

Regarding appearance, 98% (100 respondents) agreed that the bottle has an attractive design. It was thus possible to conclude that in ergonomic and visual terms, the bottle will not need changes in morphology. Concerning appearance, the following Table 3 synthesizes the questionnaire results:

**Table 3.** Usability questionnaire results concerning aspect.

| Question | Percentage | Answer |
|---|---|---|
| Are the colors on the bottle appealing? | 90.2% (92 respondents) | Agree with the visual aspects applied to the product, regardless of the used material |
| Is the stopper easy to use? | 93.1% (95 respondents) | Considered it was easy to use |
| Do you think this product meets your needs? | 96.1% (98 respondents) | Considered the product meets their everyday needs |

The participation of target users in the design validation process of a new product proved to be advantageous as it allows designers to reflect on perceived weaknesses, thus allowing the improvement of the product in a direction that will enhance purchase probabilities, subsequently allowing users to change consumption habits, avoiding single-use plastic water bottles and the corresponding plastic waste dissemination, and deleterious effects on the planet, namely on marine life.

## 5. Discussion

The concept of sustainability has been widely discussed in recent decades in all areas of society and has become an increasingly constant presence in our daily lives [1,2]. HEIs play a decisive role not only in the training of future generations of decision-makers and professionals, providing them with the specific knowledge necessary to understand the interactions between human beings and the environment [6–11] but also by promoting a smarter and more sustainable campus designed to favoring wellbeing, health and safety, waste reduction, moderating water and energy consumption, promoting local and regional community participation, and developing new curricular environmental activities. All these actions are part of HEIs' effort toward sustainable development [9,11,51]. Here it is important to clarify that the so-called sustainable design has been a tool applied to reinforce HEIs' sustainability by providing new solutions to solve old problems, similar to this particular case, the over-use of plastic in bars, canteens, and halls of residence [3,7,8]. Since sustainable design is a recent area of research, and there are still many functional, methodological, and information gaps to be filled, namely by promoting a change in the "business as usual" scenarios, where many interests are at play, and the controversy has always been a part of the process [14,16,18,19], there is still a long way to run regarding the mentalities and habits of the average citizen since sustainability requires behavior change [1,8,18]. Thus, it is urgent and essential that at an educational level, schools and universities may equip citizens, future professionals, and future decision-makers with the necessary tools for change [7,10,11,19]. By applying sustainable design tools, the Refill_H2O

project, addressed in this work not only has involved the students and teachers in the search for a new integrated system designed to replace single-use plastic bottles but has touched the whole academic fabric with hands-on collaboration, participation in decision-making processes, and real learning by having an experience with a real final system of objects with a specific shape and function, but also a common purpose and meaning with the ultimate goal of creating a more sustainable school environment [4,5,8,9,21,23].

Moreover, the sustainable design stands for the construction of meanings in a real-life problem context, like the excessive plastic consumption in the academy's daily life, allowing the construction of solutions that combine concepts related to the design, new IoT technologies, and environment protection, transcending, therefore, the classroom and the walls of the academic premises [8,10,14,16,21,24]. By developing new integrated solutions focused on reaching the higher sustainability principle of plastic waste reduction, the researchers worked toward their goals, incorporating new knowledge as they moved along [21,26,28,29]. As a consequence of this research focused on a very specific sustainability topic concerning plastic waste reduction, the results exceeded the expectations, and the engineering and design binomial was taken to a different level, where the developed objects meet requirements related to shape function, durability, usability, and sustainability, liquefying the approach of formal design and developed receiving relevant contributions from new technologies relevant to improve the final product's performance [31–35]. The sustainable design experience contributed to solidifying the theoretical and practical contents of the product design by integrating new technological concepts provided by disciplines that allowed to put into action all intervening parts that acted as agents rather than spectators, assimilating and integrating the wide range of aspects pertaining to sustainability, design, and change [13,20,28]. The transdisciplinary approach dissolved boundaries between conventional disciplines, such as Product Design, Electronics and Materials Engineering, and organized the product development around real-world problems [18] concerning a major sustainability issue: plastic waste reduction.

In order to undertake the referred transdisciplinary approach, methodology became a determining factor to answer the needs that emerged during this research process [32], which has strengthened and enriched this investigation. Thus, it was possible to optimize the project, improving it in specific stages and different aspects, thus reaching a greater potential for innovation. The interplay between creativity and engineering explains the expression of Creative Engineering relating to creative improvement and interpretation [42]. This alliance allowed the technology resulting from the sum of these areas to involve different fundamental agents for innovation in the quest for sustainability [9]. Focusing on sustainability, the strategic perspective stimulated by the Refill_H2O project is to raise awareness and motivate future actions for innovative and sustainable products, thus contributing to the sustainability of human life on the planet, which goes far beyond the development of a Smartbottle Ecosystem in a HEI [3–9]. The inclusion of ICT and IoT technologies demonstrates the meaning of the information society and responds to the aptitude of post-Millennial generations towards everyday interaction with technology. The interactive Smartbottle that communicates with an intelligent refilling station was an essential element for the enthusiasm in the reception of the system and attribution of meaning. The meaning of user-oriented sustainable design has thus become clearer by integrating these novel valences in the pursuit of sustainable solutions. Gamification is also a valuable aspect in providing the user with a memorable experience, one that is worth repeating [44,45]. If the user action is pleasant and rewarding, there are much better chances that the action will be repeated, encouraging product use, adherence, and loyalty, in an ongoing emotional relationship that boosts the Smartbottle Ecosystem use in IPVC academic environment, promoting, therefore, sustainability implementation by replacing single-use plastic bottles [51].

The survey in the academic community allowed the identification of a set of physical, aesthetic, and functional characteristics to inform the specifications of the product. Furthermore, it was possible to prove that innovation through sustainable design and new

technologies is useful and may promote systemic changes in the behavior of individuals and their communities [3]. This experience demonstrates how sustainable design may impact life, as a fundamental transformer of society, by deploying social propositions and influencing attitudes and minds in search of sustainable behaviors [6]. Materializing the axiom that human needs do not include environmental degradation, sustainable design has the power of raising public awareness, and from an ethical perspective, improving the world [19].

Since it is a novel approach, the sustainable design must refine new technical skills and critical mass to address the multiple problems arising from the imperative of sustainability [14]. Knowledge and preparation, as well as the conceptual and creative modes, are supported and reinforced by a participatory design method, adopting new research methods merging different fields of knowledge, always bearing in mind the need to rigorously, honestly, and factually address the problems to tackle unsustainability. To encourage the encounter of these two worlds (technical and creative/theoretical and practical/ academic and industrial), the sustainable design must incorporate sustainability values and standards in the current lexicon of everyone involved in the academic environment (students, professors, staff, and decision-makers) [8–10].

## 6. Conclusions

The implemented research addresses the design and implementation of an interactive smartbottle that communicates with a smart water refill station, designed to enhance the final users' enthusiasm and motivation towards environmentally friendly approaches, considering nature resources and more planet-friendly materials as part of the design process, and thus enabling the elimination of single-use water plastic bottles in a Higher Education Institution, promoting, therefore, the sustainability in the academic environment. To stimulate the sustainability mindset and ecological awareness, the smart water dispensing station was designed to display information concerning individual water intake but also environmental sustainability metrics and indicators, such as the estimated amount of averted plastic waste, the energy-saving from overall waste reduction and the reduction of greenhouse gas emission, and information on the user's environmental footprint. As a result of this investigation, three main contributions have been delivered: (i) a novel methodology based on the association of Design Thinking and Participatory Design as the basis of Sustainable Design; (ii) the design and development of an IoT-enabled smartbottle prototype; and (iii) usability evaluation of the proposed prototype.

**Author Contributions:** Conceptualization: A.F.C., A.C. and S.I.L.; methodology: A.F.C., A.C. and S.I.L.; investigation: A.F.C., J.M. and S.I.L.; writing—original draft preparation, A.F.C., J.M., A.C. and S.I.L.; writing—review and editing: A.F.C., A.C. and S.I.L.; supervision: A.F.C. and S.I.L.; project administration: A.C. All authors have read and agreed to the published version of the manuscript.

**Funding:** This research was funded by the Program Environment, Climate Change, and Low Carbon Economy, and was created following the establishment of a Memorandum of Understanding between Portugal, and Iceland, Liechtenstein, Norway under the EEA and Norway Grants 2014–2021, for the program areas of Environment and Ecosystems (PA11), and Climate Change Mitigation and Adaptation (PA13) under the scope of the project 10_SGS#1_REFILL_H20". A.C. co-authored this work within the scope of the project proMetheus, Research Unit on Materials, Energy, and Environment for Sustainability, FCT Ref. UID/05975/2020, financed by national funds through the FCT/MCTES.

**Acknowledgments:** Um agradecimento especial ao Programa Ambiente, Alterações Climáticas e Economia de Baixo Carbono, criado na sequência da assinatura do Memorando de Entendimento entre Portugal, Noruega, Islândia e Liechtenstein, tendo em vista a aplicação em Portugal do Mecanismo Financeiro do Espaço Económico Europeu (MFEEE) 2014–2021 nas áreas programáticas Ambiente e Ecossistemas (PA11), e Mitigação e Adaptação às Alterações Climáticas (PA13), pela atribuição do financiamento 10_SGS#1_REFILL_H20, selecionado no âmbito do Aviso Small Grants Scheme #1–Projetos para a prevenção e sensibilização para a redução do lixo marinho. Este Projeto contribui para a execução do Objetivo n.º 1 do 'Programa Ambiente': "Aumentar a aplicação dos princípios da Economia Circular em sectores específicos", e do Output 1.3 do Programa, através de promoção da

Economia Circular pela "Redução de plásticos nos Oceanos, de origem em atividades terrestres", em conformidade com o Anexo I do Acordo de Programa assinado a 27 de maio de 2019.

**Conflicts of Interest:** The authors declare no conflict of interest.

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
