# Peer review of "Joining Sustainable Design and Internet of Things Technologies on Campus: The IPVC Smartbottle Practical Case"

_sustainability, doi:10.3390/su14105922_

Round 1
Reviewer 1 Report
Dear authors, the presented article, its clear and well written. The topic its relevant with the exemplification of a product design case in the educational comunity using digital, technologies, semantics, design methods and user centered design. It seems it fits in the topics of this journal, but personally I dont see much of a contribution in scientific terms, the design methods and developed proposals are not new, but still interesting and motivating for a better future.
I recommend to rewrite the abstrac as it is no clear. and include more references that give an stronger theoretical background to the paper.
Author Response
We would like to thank Reviewer 1 for his valuable contribution. Although fitting the topics of the journal, the contribution in scientific terms was not clear, and thus this issue was developed. As advised, the abstract was rewritten, and further bibliography was included concerning the theoretical background, even though the design methods and proposals are not new, as duly noted. They are nonetheless actual, are up to date in use and have been widely proven effective.

Reviewer 2 Report
This article presents the project development of a sustainable smart bottle that tracks users’ consumption of water throughout the day. The bottle is connected to an app. The research focuses on the co-design process to develop the bottle (product) and system to track consumption. The article is well written, scientifically sound and interesting to read. But, I would not recommend it for publication in Sustainability. The main reason is that the product development is not ‘sustainable’ if we think of the context in which that product is developed. The authors emphasize on the fact that the bottle is ‘sustainable’ in how it would be manufactured and that it is more sustainable than using non reusable bottles. I agree that reusable bottles are more sustainable that non reusable bottle but, as the study shows, the future users targeted (faculty, students on campus) that were interviewed as part of the project development already use their reusable bottles. 412 out of 536 respondents already use a reusable bottle (that’s more than 75%). The most sustainable solution would be for users to keep using their bottles and to find a solution to adapt their bottles to become ‘smart’ in order to track water consumption etc. To produce new bottles to replace working bottles means producing more and aligns with a consumerism approach that is not sustainable in itself. Unless the author can justify that it is sustainable to produce new reusable bottle to replace old reusable bottles for 75% of their target users, I don’t think that emphasizing the sustainability of the product is the way to go for the research presented.
References I think could benefit the author on that topic are:
McDonough, W., & Braungart, M. (2002). Cradle to cradle: Remaking the way we make things. North Point Press.
Cucuzzella, C. (2016). Creativity, sustainable design and risk management. Journal of Cleaner Production, 135, 1548–1558. https://doi.org/10.1016/j.jclepro.2015.12.076
Cradle to cradle is a great book to reflect on how we make / produce things. The Cucuzella paper tackle the levels on innovation in sustainability when applying to product development. It explains through examples how higher levels of innovation can really have a sustainable impact.
I think a design journal would be more appropriate to publish the research. As I said above, I think the paper is well-written, the topic is interesting. To me, the participatory approach to designing is more relevant to emphasize than the sustainability part that I found flawed. I think the paper should focus on the challenges on developing a product through participatory design, while including sustainability as a characteristic of the end product. The authors make interesting points along the paper on user-oriented design (p.11 line 439). I think the paper should be reoriented on the design process itself, and the sustainable part be an example of a requirement of the design brief.
Here are comments that could help improve the paper further:
I suggest to include sub sections in the introduction for the reader to better grasp the main concepts / ideas developed.
The objective of the paper is stated in p.3 line 104 to 108. The objective is only descriptive about the development of a product. What is the bigger picture? As a reader what can I get from it that is different than other research works on that topic? The paper falls short into ‘zooming out’ of the case study presented. The case study should serve as input to discuss ‘bigger’ questions like participatory design or sustainable design. What were the challenges from the project development? How does it align or misalign with Brown’s (or others) model of Design Thinking? Is the model presented on Fig.1 a result? If so, that would be the contribution of the paper and the authors should connect how the results of the case study fit into that model. Then, how does this model differ from previous ones? What are other models of sustainable product development? Where those models flawed which is why the authors are presenting their model?
Typos:
Abstract line 27: simultaneous > Simultaneous (need caps on ‘s’)
p.2 line 57: need a ‘.’ Between ‘years’ and ‘Orcas’
Author Response
We would like to thank Reviewer 2 for his careful and insightful contribution. The identified typos have been corrected.

Round 2
Reviewer 1 Report
No more comments.
Author Response
The authors wish to thank the Reviewer his time and dedication to improve our article, as well as the thoughtful consideration of useful elements for a better text organization. The suggested changes were implemented and are highlighted in red font.

Reviewer 2 Report
I appreciate the authors' efforts in their revision of the paper. I enjoyed reading the introduction that sets the motivation and objectives of the paper. I have more comments to improve the paper, mainly concerning the background section and the discussion that should be added.
There are a lot of statements in the background section that are not referenced. I would add a few references where relevant either from the authors work or other research to support those statements.
In the background section, there are three paragraphs that define the current case study of Refill H20. I am not convinced that this is the best place for it as the background should be more general about the 'setting' and context for your research. Those should be moved somewhere else, probably at the end of the method to present Refill H20 as a case study joining sustainable design and internet of things technology. It would be interesting to have examples of similar projects, though. Are there other examples of projects or actions similar to Refill H20 that could be referenced in the background? What were their implementations? What was their design process? How is your case study / project different? Similar?
The background will benefit in clarity by having two subsections, one on sustainable design / sustainable development (the current few paragraphs + examples of analogous projects to Refill H20) and another subsection on the design thinking / design method framework that includes participatory design and Figure 1. The authors state that they contribute to both topics so we need a more precise setting of what already exists (with references from others, Tim Brown and more) to assess the scientific contribution of the current study.
There is a lot of information in the background section but the transition between ideas is not that fluid and should be improved. Overall, the background needs a bit of restructuring.
The author should integrate a discussion section. The section called Conclusion is more of a discussion. I suggest to name it discussion and add a conclusion section with a ten-line paragraph that summarizes the motivation, the study, the main results / findings, the challenges raised by the research and future direction for that research topic in general and your work in particular.
For the discussion section, currently called Conclusion, I suggest to reorganize it so that it addresses the three contributions outline in the introduction : i) proposal of a novel methodology based on the fusion of Design Thinking and Participatory Design; ii) the design and development of an IoT enabled smart bottle prototype; and iii) usability evaluation of the proposed prototype. For each of those, the authors need to discuss their work in perspective with what others have done in past research so that it highlights the contributions and limitations of the results presented. There are elements of discussion in the conclusions about each of the three contributions cited in the intro but there is a lack of reference to other work that should be added. What did the framework applied added to the design process? Why is it relevant for sustainable design based on your experiment? compared to other experiments? What is the specificity of your approach to joining sustainable design and IOT compared to previous projects? What is your contribution to usability evaluation from a research perspective, not only in terms of results? Maybe a more efficient way would be to reframe the contribution. From reading the article, I would suggest to simplify the contribution to the process (participatory design and extension / testing of Brown's model) and the product (sustainable design and IOT). The case study provides insights on both and those insights should be discussed in relation to other research done on these topics.
One last comment, what were the limitations of this study? Possible bias? And how would you address it in future work? Is there a second phase to the project where you’ll evaluate the 'success' of the refill station?
Author Response

(The authors gave the same response as above.)

Round 3
Reviewer 2 Report
I enjoyed reading the paper. I suggest for it to be accepted for publication after these few minor comments are included:
Please provide a more descriptive section title for 2.2 Case studies. It would help a reader that scan through the paper to get better information about the content. Maybe Case studies of smart bottle projects or Case studies similar to Refill H20 or Case studies of product/actions to reduce the use of plastic water bottles. In this section, maybe add a sentence to smooth the transition from the previous section. At the end of the section, please conclude on the section. Something like: Through these examples similar to Refill H20, we see that xxxx so it shows that this topic is important because xxxx . This is why our study is important etc. Now we will present the methodology used to conduct our experiment / analysis / implementation of Refill H20 etc. I think this new section is great to set the stage but needs smother connections to what is before and after.
Suggestion for language change / typo:
Line 28 fusion > association
Line 102 alliance > association
Line 167 replace . by , after [18]
Line 211 adopted > adapted
Line 245 delete ‘multiple dimensions to consider regarding’ to avoid repetition from line above
Line 297-298 Please reformulate “However, it is essential to emphasize that the two work these steps” My understanding is that the engineer and the designer go through same steps but with different approaches. Can you reformulate it in a clearer way?
Author Response

(The authors gave the same response as above.)
